# Agreement between physician- and patient-reported Canadian cardiovascular society scores among patients undergoing elective coronary angiography–The CATS study

Piia Lavikainen [1,2]*, Juha Hartikainen[1,3], Heikki Miettinen[1], Marketta Viljakainen[1], Janne Martikainen[2], Anna-Maija Tolppanen[2], Risto P. Roine[1,4]

1 Heart Center, Kuopio University Hospital, Kuopio, Finland, 2 School of Pharmacy, University of Eastern Finland, Kuopio, Finland, 3 School of Medicine, University of Eastern Finland, Kuopio, Finland, 4 Department of Health and Social Management, University of Eastern Finland, Kuopio, Finland

* piia.lavikainen@uef.fi

## Abstract

The primary aim of revascularization in stable coronary artery disease (CAD) is symptom relief. The severity of symptoms is usually evaluated by the physician, not by the patient. We examined the agreement between physician- and patient-reported Canadian Cardiovascular Society (CCS) scores among patients scheduled for elective coronary angiography in a cross-sectional study. Patients (n = 650) and cardiologists evaluated the severity of angina symptoms by filling the CCS questionnaire before coronary angiography. Patients were divided into those without CAD (stenosis diameter <50%, n = 445) and those with CAD (stenosis diameter >50%, n = 205). CAD patients were further divided into three groups according to disease severity (single-, double- or triple-vessel disease). The mean age of the patients was 67.6 (9.9) years and 50.6% were women. In 51.8% (95% CI 44.5%–59.0%) of patients with CAD and 51.9% (95% CI 47.0%–56.8%) of those without, physician- and patient reported CCS scores agreed. The physician reported better CCS scores in 33.9% (95% CI 27.6%–40.7%) of patients with CAD and 36.2% (95% CI 31.8%–41.0%) of patients without CAD. The proportions of full or partial agreement between physician- and patient reported CCS scores were similar across the CAD severity groups. To summarize, we observed a significant discrepancy between the physician- and patient-reported symptom severity in patients with or without CAD scheduled for angiography. The physician underestimated the symptoms in third of the cases. Thus, patient-reported symptom severity, rather than physician's evaluation, should be the cornerstone of treatment decisions.

## Introduction

The primary indication for revascularization in stable coronary artery disease (CAD) is symptom relief. Therefore, the severity of symptoms is crucial in guiding treatment decisions and in evaluating treatment effect [1]. Canadian Cardiovascular Society (CCS) grading is the most

**Data Availability Statement:** Access to data is regulated by Finnish law. Data are available from

the Kuopio University Hospital for researchers who meet the criteria as required by the Finnish law for access to confidential data. Contact person who will distribute data upon request to qualified researchers: Juha Hartikainen, Heart Center, Kuopio University Hospital, PO BOX 100, KYS FI-70029, Finland; juha.hartikainen@kuh.fi.

**Funding:** This research was supported by the State Clinical Research Fund (VTR) of Kuopio University Hospital, Kuopio, Finland in the form of a salary to MV [diary number 13.11.2014/19§]. The specific roles of these authors are articulated in the 'author contributions' section. The funder had no role in the design and conduct of the study; collection, management, analysis, and interpretation of the data; preparation, review, or approval of the manuscript; and the decision to submit the manuscript for publication.

**Competing interests:** The authors have read the journal's policy and have the following competing interests: Prof. Janne Martikainen is a founding partner of ESiOR Oy. This company was not involved in carrying out this research. This does not alter our adherence to PLOS ONE policies on sharing data and materials. There are no patents, products in development or marketed products associated with this research to declare.

used tool to describe the severity of angina pectoris symptoms [2]. However, CCS is usually assessed by the treating physician, not by the patient. This carries an inherent risk for bias, particularly if the effectiveness of treatment is addressed. To overcome this bias, patient-reported outcomes (PROMs) should be used for symptom evaluation [3, 4].

A recent study reported 36% agreement between physician- and patient-reported CCS scores in patients with multi-vessel CAD undergoing revascularization with CCS scores estimated after the coronary angiography [5]. We examined the agreement between physician- and patient-reported CCS scores among patients undergoing diagnostic coronary angiography and, thus, unaware of the result of the angiography.

## Materials and methods

Altogether 650 patients scheduled for diagnostic coronary angiography (NOMESCO Classification of Surgical Procedures codes FN1AC, FN1BB, FN1BC, and FN1CC) at the Kuopio University Hospital, Finland between VI/2015 and VI/2017 filled in the CCS questionnaires (see S1 File) prior to angiography. Physicians performing the angiography evaluated patients' CCS class before the procedure while being unaware of the patient-evaluated CCS score.

Patients were divided into those without significant CAD (stenosis diameter <50%) and those with CAD (stenosis diameter ≥50%). Those with CAD were further grouped according to the number of major coronary arteries with significant stenosis (single-, double- or triple-vessel disease).

The study was approved by the Research Ethics Committee of the Northern Savo Hospital District (numbers 3/2014, 221/2016, and 135/2017). Written informed consent was obtained from the patients as required by the Ethics Committee. The authors used only pseudonymized data.

### Outcomes

Both physician- and patient-reported CCS scores were recorded with four-class accuracy [2]. Class I represents angina only in strenuous exertion, class II angina in moderate and class III in mild exertion. In class IV, angina symptoms may be present at rest.

### Statistical analyses

Differences in patient characteristics were compared with Student's t-tests and $\chi^2$-test. Differences in physician- and patient-reported CCS scores were tested with paired samples t-test.

The agreement between physician- and patient-reported scores was evaluated with Spearman correlation and by classifying the difference in scores as full agreement, or partial (one-point or two-point) disagreement. All the comparisons between physician- and patient-reported CCS scores were conducted among patients, whose both scores were available. Statistical analyses were performed with SAS 9.4 (SAS Institute Inc., Cary, North Carolina).

## Results

Coronary angiography revealed significant CAD in 205 of 650 (31.5%) patients (Table 1). Of them, 106 (51.7%) had single-vessel, 56 (27.3%) double-vessel and 43 (21.0%) triple-vessel diseases. Patients with CAD were older, more often men, and had a worse physician-reported mean CCS score than patients without CAD (Table 1, Fig 1). Mean physician-reported CCS score was worse in patients with triple-vessel disease compared to those with single-vessel disease. Patients with triple-vessel disease had on average worse mean self-reported CCS score than patients with single- or double-vessel disease or those without CAD.

**Table 1. Characteristics of the study population.**

| | Total population | Without CAD | With CAD | *Single-vessel disease* | *Double-vessel disease* | *Triple-vessel disease* |
|---|---|---|---|---|---|---|
| | n = 650 | n = 445 | n = 205 | n = 106 | n = 56 | n = 43 |
| **Age, y** | 67.6 (9.9) | 66.4 (10.06) | 70.2 (8.81) *** | 69.4 (8.52) # | 70.1 (9.78) | 72.4 (7.96) |
| **Female** | 329 (50.6) | 247 (55.9) | 82 (40.0) *** | 55 (51.9) ## | 17 (30.4) | 10 (23.3) |
| **Physician-reported outcome:** | | | | | | |
| **CCS** | 1.98 (0.59) | 1.93 (0.59) | 2.07 (0.59) ** | 2.00 (0.53) # | 2.11 (0.62) | 2.21 (0.64) |
| I No limitation of physical activity | 114 (17.5) | 88 (19.8) | 26 (12.7) | 15 (14.2) | 7 (12.5) | 4 (9.3) |
| II Slight limitation of physical activity | 438 (67.4) | 298 (67.0) | 140 (68.3) | 76 (71.7) | 37 (66.1) | 27 (62.8) |
| III Marked limitation of physical activity | 86 (13.2) | 49 (11.0) | 37 (18.0) | 15 (14.2) | 11 (19.6) | 11 (25.6) |
| IV Unable to carry on any physical activity without discomfort | 6 (0.9) | 4 (0.9) | 2 (1.0) | 0 | 1 (1.8) | 1 (2.3) |
| missing | 6 (0.9) | 6 (1.3) | 0 | 0 | 0 | 0 |
| **Patient-reported outcome:** | | | | | | |
| **CCS** | 2.33 (0.82) | 2.30 (0.81) | 2.38 (0.83) | 2.29 (0.83) ## | 2.31 (0.76) ## | 2.69 (0.87) |
| I No limitation of physical activity | 64 (9.8) | 44 (9.9) | 20 (9.8) | 14 (13.2) | 4 (7.1) | 2 (4.7) |
| II Slight limitation of physical activity | 359 (55.2) | 254 (57.1) | 105 (51.2) | 54 (50.9) | 33 (58.9) | 18 (41.9) |
| III Marked limitation of physical activity | 119 (18.3) | 73 (16.4) | 46 (22.4) | 23 (21.7) | 10 (17.9) | 13 (30.2) |
| IV Unable to carry on any physical activity without discomfort | 73 (11.2) | 49 (11.0) | 24 (11.7) | 10 (9.4) | 5 (8.9) | 9 (20.9) |
| missing | 35 (5.4) | 25 (5.6) | 10 (4.9) | 5 (4.7) | 4 (7.1) | 1 (2.3) |

**Notes:** Statistical significances

* p<0.05

** p<0.01, and

*** p<0.001 vs. without CAD

# p<0.05

## p<0.01, and

### p<0.001 vs. triple-vessel disease.

**Abbreviations:** CAD, coronary artery disease; CCS, Canadian Cardiovascular Society. The values are No. (%) or mean (standard deviation).

When comparing physician- and patient-reported mean CCS scores, the physician-reported CCS score was lower (less severe) than the patient-reported CCS score (Fig 1). This was true both for patients with or without CAD and for patients with single- or triple-vessel disease. The physician- and patient reported CCS scores agreed fully for only 51.8% (95% CI 44.5%–59.0%) of patients with CAD and for 51.9% (95% CI 47.0%–56.8%) of those without CAD (Table 2). For 33.9% (95% CI 27.6%–40.7%) of the patients with CAD and 36.2% (95% CI 31.8%–41.0%) of the patients without CAD, the physician reported at least one-point better CCS score than the patient. On the contrary, the physicians reported at least one-point worse CCS score than the patient in 14.4% (95% CI 10.1%–20.0%) of patients with CAD and 11.8% (95% CI 9.1%–15.3%) of those without CAD.

The correlation coefficients between physician- and patient-reported CCS scores were low, 0.22 (95% CI 0.09–0.35) and 0.24 (95% CI 0.15–0.33) in patients with and without CAD, respectively. The proportions of full or partial agreement between physician- and patient reported CCS scores were similar across the CAD severity categories (Table 2).

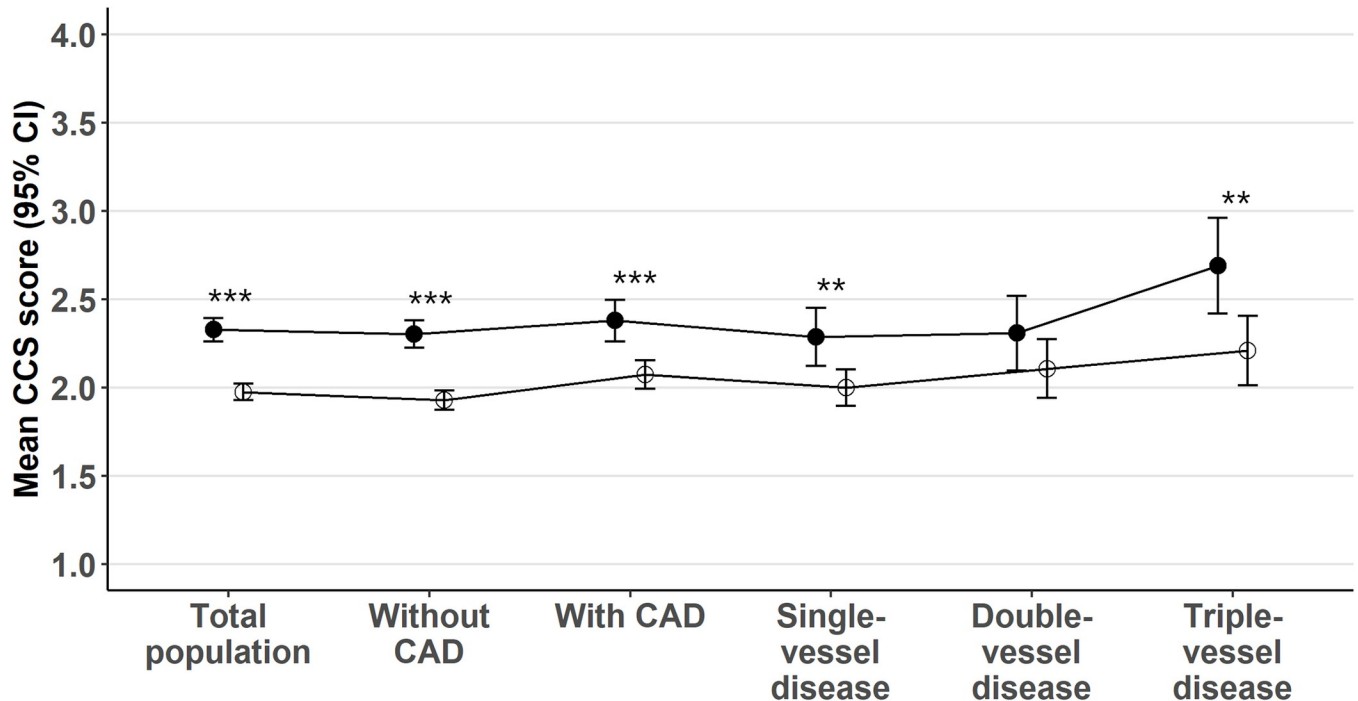

**Fig 1. Patient- and physician-reported CCS scores.** Statistical significances: * p<0.05, ** p<0.01, and *** p<0.001 for patient- vs. physician-reported CCS.

## Discussion

We demonstrated a significant discrepancy in the severity of angina symptoms assessment between the physician and the patient. The physician- and patient-reported CCS grading agreed only in half of the cases. The physicians underestimated symptoms in one out of three patients scheduled for diagnostic coronary angiography. Similar discrepancies were observed among patients with and without CAD, and regardless of CAD severity.

**Table 2. Agreement between patient- and physician-reported CCS scores.**

| | Total population | Without CAD | With CAD | *Single-vessel disease* | *Double-vessel disease* | *Triple-vessel disease* |
|---|---|---|---|---|---|---|
| | n = 609 | n = 414 | n = 195 | n = 101 | n = 52 | n = 42 |
| **Full agreement** | 51.9 (47.9–55.8) | 51.9 (47.0–56.8) | 51.8 (44.5–59.0) | 53.5 (43.3–63.5) | 53.8 (39.5–67.8) | 45.2 (29.8–61.3) |
| **Physician-reported CCS one point better than patient-reported** | 23.6 (20.4–27.2) | 23.9 (19.9–28.3) | 23.1 (17.4–29.6) | 23.7 (15.9–33.3) | 21.2 (11.1–34.7) | 23.8 (12.1–39.5) |
| **Physician-reported CCS at least two points better than patient-reported** | 11.8 (9.5–14.6) | 12.3 (9.5–15.8) | 10.8 (7.2–15.9) | 9.9 (4.9–17.5) | 7.7 (3.0–18.2) | 16.7 (8.0–30.6) |
| **Physician-reported CCS one point worse than patient-reported** | 11.7 (9.3–14.5) | 11.1 (8.3–14.5) | 12.8 (8.5–18.3) | 10.9 (5.6–18.7) | 15.4 (6.9–28.1) | 14.3 (5.4–28.5) |
| **Physician-reported CCS at least two points worse than patient-reported** | 1.0 (0.5–2.1) | 0.7 (0.1–2.1) | 1.5 (0.3–4.4) | 2.0 (0.2–7.0) | 1.9 (0.0–10.3) | 0.0 (-) |

**Abbreviations:** CAD, coronary artery disease; CCS, Canadian Cardiovascular Society. The values are % (95% confidence interval).

The main purpose of stable CAD treatment is angina relief. Thus, the patient's assessment of symptoms is crucial when considering the therapeutic options. However, the symptom severity is evaluated and interpreted by the physician. The agreement between physician- and patient-rated CCS (51.9%) in our study is in line with Kemp et al., who reported 36.3% agreement between the physician- and patient-reported CCS scores in patients with multi-vessel CAD and undergoing revascularization [5]. However, in that study the physicians overestimated the symptoms in 37.8% of patients, whereas in our study the physicians underestimated the symptoms in 35.4% of patients. Importantly, in that study [5], both the physician and the patients were aware of the presence of multivessel CAD, and the treatment decision had already been made. By contrast, in our study the CCS scores were evaluated before angiography when both the physician and the patient were unaware of the angiography result.

The awareness of the presence of CAD and the need for revascularization may influence the assessment of symptom severity, particularly by the physician. In a study comparing physician- and patient-estimated angina in patients undergoing percutaneous coronary intervention (PCI) 35% of patients undergoing elective PCI and 12% of those scheduled for urgent PCI reported no angina preoperatively [6]. However, the physician estimated that 20% of the symptom-free patients scheduled for elective PCI and 12% of those scheduled for urgent PCI had moderate to severe angina.

Our results agree also with the APPREAR study [7, 8]. It compared patient-reported severity of angina using the Seattle Angina Questionnaire (SAQ) and physician-assessed angina severity utilizing structured questions on, for example, angina frequency among CAD patients in cardiology outpatient clinics. The physician missed 42% of patients reporting angina during previous month [7], and of patients with daily or weekly angina, 26% were assessed as being free of angina by the physician [8]. However, these studies applied SAQ, typically used for research, not CCS which is the most used angina scoring tool in clinical practise.

The agreement between physician- and patient-estimated CCS score in our study (52%) is close to the agreement (54%) reported between two cardiologists assessing symptom severity among heart failure patients using the New York Heart Association (NYHA) grading [9]. The disagreement between the physicians was related to problems in distinguishing patients belonging to NYHA classes II and III. In our study, the disagreement between patient- and physician-reported symptom severity was evident in all CCS classes and most pronounced in the extremities of the disease severity scale (CCS classes I and IV). In our study, 18% of physicians, but 10% of patients classified symptom severity as mild (class I), whereas angina at rest (CCS class IV) was reported by 11% of the patients, but only by 1% of the physicians.

The differences between patient- and physician-evaluated CCS scores may partially arise from possible limitations of CCS related to inter- and intra-rater reproducibility, content validity and lack of sensitivity to differentiate between all levels of angina symptoms [10]. Still, our results demonstrate an important discrepancy between the two ratings and highlight that the symptoms should be evaluated by the patient herself/himself, not by the physician.

U.S. Food and Drug Administration has stated that assessment of patients' health and symptoms should come directly from the patient without interpretation by a clinician or a healthcare provider [11]. The added value of PROMs is that they support patient-provider engagement by assessing the severity of symptoms, providing additional means for tracking the effect of treatments, informing treatment decisions, and monitoring general health and well-being. Further, the process of self-reporting *per se* can improve symptom management, quality of life, communication, and satisfaction with care [4, 12]. Therefore, patient-reported symptom severity should be more efficiently incorporated into practice.

A strength is that the CCS scores were recorded before angiography. Thus, the result of the invasive examination did not influence the assessment of symptom severity.

Selection bias is a potential limitation of our study. Only patients with both physician- and self-reported CCS scores were included. The response rate in an earlier study in patients scheduled for elective angiography and PCI in the Kuopio University Hospital was circa 40% [13]. Typically, the most comorbid and oldest patients are least prone to answer surveys [12]. However, this does not impact the conclusion of our study.

## Conclusion

To conclude, our results show that physicians are prone to underestimate patients' symptoms. When the primary aim of treatment is symptom relief, as it is in stable CAD, self-reported symptom assessment, rather than physician's evaluation, should be the cornerstone of treatment decisions.

## Supporting information

**S1 File. The CCS questionnaire.**
(PDF)

## Author Contributions

**Conceptualization:** Piia Lavikainen, Juha Hartikainen, Heikki Miettinen, Janne Martikainen, Anna-Maija Tolppanen, Risto P. Roine.

**Data curation:** Marketta Viljakainen.

**Formal analysis:** Piia Lavikainen.

**Funding acquisition:** Juha Hartikainen, Anna-Maija Tolppanen, Risto P. Roine.

**Investigation:** Piia Lavikainen.

**Methodology:** Piia Lavikainen.

**Project administration:** Juha Hartikainen, Marketta Viljakainen, Risto P. Roine.

**Resources:** Juha Hartikainen, Marketta Viljakainen.

**Software:** Piia Lavikainen.

**Supervision:** Juha Hartikainen, Heikki Miettinen, Janne Martikainen, Anna-Maija Tolppanen, Risto P. Roine.

**Validation:** Anna-Maija Tolppanen.

**Visualization:** Piia Lavikainen.

**Writing – original draft:** Piia Lavikainen, Juha Hartikainen, Anna-Maija Tolppanen, Risto P. Roine.

**Writing – review & editing:** Piia Lavikainen, Juha Hartikainen, Heikki Miettinen, Marketta Viljakainen, Janne Martikainen, Anna-Maija Tolppanen, Risto P. Roine.

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
