## [Decision Letter · Decision Letter 0]

13 Sep 2023

Agreement between physician- and patient-reported Canadian Cardiovascular Society scores among patients undergoing elective coronary angiography – The CATS study

PONE-D-23-17358

Dear Dr. Lavikainen,

We’re pleased to inform you that your manuscript has been judged scientifically suitable for publication and will be formally accepted for publication once it meets all outstanding technical requirements.

Kind regards,

Gorica Maric

Academic Editor

PLOS ONE

Journal Requirements:

1. Thank you for stating in your Funding Statement: 

"This research was supported by the State Clinical Research Fund (VTR) of Kuopio University Hospital, Kuopio, Finland [diary number 13.11.2014/19§]."

Please respond by return e-mail so that we can amend your financial disclosure and competing interests on your behalf.

" Prof. Janne Martikainen is a founding partner of ESiOR Oy and a board member of Siltana Oy. These companies were not involved in carrying out this research. No other conflicts of interest were reported."

Please respond by return email with your amended Competing Interests Statement and we will change the online submission form on your behalf.

Additional Editor Comments (optional):

Reviewers' comments:

Reviewer's Responses to Questions

**Comments to the Author**

1. Is the manuscript technically sound, and do the data support the conclusions?

Reviewer #1: Yes

Reviewer #2: Yes

2. Has the statistical analysis been performed appropriately and rigorously? 

Reviewer #1: Yes

Reviewer #2: Yes

3. Have the authors made all data underlying the findings in their manuscript fully available?

Reviewer #1: Yes

Reviewer #2: Yes

4. Is the manuscript presented in an intelligible fashion and written in standard English?

Reviewer #1: Yes

Reviewer #2: Yes

5. Review Comments to the Author

Reviewer #1: This paper has addressed a fundamental aspect of treating CAD/ and angina that is based on symptomatic treatment. It rises the issue of physicians reaching medically sound decisions based on their medical knowledge , while taking into consideration the patients evaluation of their symptoms.

In clinical practice there will always be a spectrum of symptoms being graded differently from one patient to the other. It is difficult quantifying feeling But the ethically sound decision would be to trust the patients complain and try to exhaust all available resources to provide the best care possible.

Several studies across the different departments of medicine have highlighted the issue of patient vs physician evaluation of pain/symptoms.

The following issues arise with this paper:

1-As mentioned int he manuscript, the issue of selection bias arises. However the overall outcome is reasonable and satisfactory in my opinion.

2-In the CCS class questionnaire, the first few lines explaining the symptoms associated with angina, may have lead to some patients responding in a more exaggerated manner, or lead to recall bias.

Finally, the manuscript does address an issue previously addressed in other areas, and definitely needs to be assed in cardiology.

Despite the potential for bias afore mentioned, the manuscript show strong evidence supporting th hypothesis.

Reviewer #2: The authors presented th.e reported CCS scores by patients and physician before coronar angiography. The authors could underline that physcian have a trend to underestimate the severity as measured with the CCS regarding treatment decision. As the relief from symptoms of ischemia is the main aim in stable coronary artery disease the application of patient reported CCS might be an alternativ to receive better results regarding the treatment success. The physician reported outcomes might not be as accurate and are often too inaccurate to evaluate the outcome of the patients.

The study is well designed and all limitations were addressed in the text. Tables and figures are as well good in the manuscript.

6. PLOS authors have the option to publish the peer review history of their article (what does this mean?). If published, this will include your full peer review and any attached files.

Reviewer #1: **Yes: **Maha Ahmed

Reviewer #2: **Yes: **Christoph Sinning

---

## [Editor Report · Acceptance letter]

5 Oct 2023

PONE-D-23-17358 

Agreement between physician- and patient-reported Canadian Cardiovascular Society scores among patients undergoing elective coronary angiography – The CATS study 

Dear Dr. Lavikainen:

I'm pleased to inform you that your manuscript has been deemed suitable for publication in PLOS ONE. Congratulations! Your manuscript is now with our production department. 

Kind regards, 

on behalf of

Dr. Gorica Maric 

Academic Editor

PLOS ONE